# Influence of bodyweight on prednisolone pharmacokinetics in dogs

**Bonnie L. Purcell**[1,2]**, Andrew P. Woodward**[3,4]**, Michael G. Leeming**[5]**,
Julien Rodolphe Samuel Dandrieux**[1,6]*

**1** Department of Veterinary Clinical Sciences, University of Melbourne, Werribee, Victoria, Australia,
**2** Peninsula Vet Emergency and Referral Hospital, Mornington, Victoria, Australia, **3** Faculty of Health,
University of Canberra, Bruce, Australian Capital Territory, Australia, **4** Precision One Health Initiative,
University of Georgia College of Veterinary Medicine, Athens, Georgia, United States of America,
**5** Melbourne Mass Spectrometry and Proteomics Facility, Molecular Science & Biotechnology Institute,
The University of Melbourne, Parkville, Victoria, Australia, **6** Royal (Dick) School of Veterinary Studies,
University of Edinburgh, Easter Bush, United Kingdom

* J.Dandrieux@ed.ac.uk

## Abstract

### Background

Larger dogs may be at greater risk of prednisolone side effects, yet there is
limited research about how bodyweight affects prednisolone pharmacokinetics in
dogs.

### Hypothesis/objectives

To describe the relationship between prednisolone dose, bodyweight, body surface
area (BSA) and prednisolone area under the curve (AUC) in dogs receiving predniso-
lone for medical reasons.

### *Animals*

25 client owned dogs receiving prednisolone for medical reasons.

### *Methods*

Observational population pharmacokinetic study. Liquid chromatography tandem
mass spectrometry was used for plasma prednisolone quantification. Data analy-
sis was conducted in a two-stage approach using non-compartmental modelling. A
Bayesian non-linear regression model described the relationship between AUC over
8 hours ($AUC_{8hr}$,ng·min/mL), bodyweight and prednisolone dose.

### *Results*

From the allometric scaling model of the form $AUC_{8h} = A \cdot BW^B$, the scaling expo-
nent $B$ was.83 (90% credible interval (CrI):.60–1.06) and the coefficient $A$ was

journal.pone.0326586

College of Veterinary Medicine and Biomedical
Sciences, UNITED STATES OF AMERICA

**Peer Review History:** PLOS recognizes the
benefits of transparency in the peer review
process; therefore, we enable the publication
of all of the content of peer review and
author responses alongside final, published
articles. The editorial history of this article is
available here: https://doi.org/10.1371/journal.
pone.0326586

**Data availability statement:** All relevant data for this study are publicly available from the GitHub repository (https://github.com/APWoodward/prednisolone_PK).

**Funding:** Financial support for this research was granted by The American College of Veterinary Internal Medicine Resident Research Grant (reference 775788). The funders had no role in study design, data collection and analysis, decision to publish, or preparation of the manuscript.

**Competing interests:** The authors have declared that no competing interests exist.

22.8 (90% CrI: 11.8–43.4). This model suggests that equivalent exposure would be obtained using an intermediate strategy between BSA and bodyweight dosing, but the total evidence provided was relatively weak.

### Conclusions and clinical importance

Evidence was obtained regarding the nonlinear relationship between prednisolone pharmacokinetics and bodyweight in dogs; however, this model is currently too imprecise for clinical dose determination.

---

## Introduction

Prednisolone is a glucocorticoid commonly used in canine medicine. It has a range of indications, including physiologic cortisol replacement, as an anti-inflammatory, and immunosuppressive agent [1]. Unfortunately, it also has a range of potential adverse effects such as polyphagia, polydipsia, polyuria, sarcopenia, gastrointestinal bleeding and increased risk of secondary infections [2–5]. Concern for side effects is greater for large dogs treated with glucocorticoids. In dogs treated with immunosuppressive doses of glucocorticoids there is a 30% increase in odds of developing muscle atrophy and polyphagia for every 5 kg of bodyweight [3]. Expert guidelines for the treatment of immune mediated hemolytic anemia recommend dose determination for dogs over 25 kg based on body surface area (BSA), rather than bodyweight, due to concern for increased side effects [6]; this recommendation rests on substantial assumptions regarding the relationship between prednisolone pharmacokinetics and body size, which are supported by little quantitative evidence.

BSA in dogs may be predicted from bodyweight by the exponential function: [7]

$$BSA = 0.101 \times BW^{2/3} \tag{1}$$

Where $BSA$ is body surface area (m²), and $BW$ the bodyweight (kg). Compared to dose determination by bodyweight, dose determination by BSA results in a proportionally decreasing dose with increasing bodyweight. However, the effect on the resulting drug exposure is dependent on the relationship between bodyweight and pharmacokinetics, which for prednisolone is not well understood. Nam and colleagues compared prednisolone concentration-time profiles from a group of beagles (bodyweight 9.9 ± 1.7 kg) to a group of large, mixed breed dogs (bodyweight 29 ± 2.9 kg) [8]. After 2 mg/kg prednisolone the larger dogs had higher maximum concentration and area under the curve (AUC) than the beagles [8]. Though supportive of non-linear effect of bodyweight on pharmacokinetics, this type of evidence cannot support quantitative decision-making for dose determination.

Similarly to BSA, physiologic processes, such as metabolic rate, do not increase linearly with increasing body size, which is captured by a theoretical framework based on the geometry of living organisms [9]. PK processes are ultimately dependent on physiology, so key PK parameters including half-life, volumes of distribution,

and clearance generally depend non-linearly on bodyweight [10]. The general theoretical support for simple mathematical relationships between bodyweight and PK suggests the feasibility of statistical models capturing these effects. For dogs, which vary in bodyweight over an order of magnitude, such a model could have substantial value in dose determination and evaluation.

The aims of this study were: (1) observe prednisolone pharmacokinetics in dogs receiving prednisolone for medical reasons. (2) describe the relationship between prednisolone dose, bodyweight, and AUC (3) implement and evaluate a statistical model for dose optimization based on bodyweight.

## Materials and methods

### Animals

Eligible subjects were dogs receiving oral prednisolone for any indication, at least 12 months of age, at least 2.5 kg bodyweight, and amenable to handling including intravenous catheter placement without requiring sedation. Exclusion criteria included use of topical prednisolone, clinically apparent liver disease (no evidence of synthetic failure on routine biochemistry or normal bile acid stimulation test), and concurrent ketoconazole or rifampicin.

Approval was obtained from the institutional animal ethics committee (Animal ethics ID 10364, The University of Melbourne). Each dog had its signalment recorded, was weighed on the day of participation, body condition and muscle condition scored. All major medical conditions and concurrent medications were recorded.

### Procedures

Each dog was presented by its owner on the morning of participation. At this time, consent and admit paperwork was performed, including confirming when the dog last received any medications. Dogs could not have received any prednisolone on the day of presentation. Dogs were not required to have food withheld prior to arriving at the hospital.

An intravenous catheter was placed for blood collection. This catheter was used for the entire sampling process providing it remained patent. If an intravenous catheter could not be placed, or became non-patent during sample collection, direct venepuncture was used but with a limited sampling protocol (described later). Prior to prednisolone administration, a baseline blood sample was taken. After this sample, each dog was administered its normal dose of prednisolone orally by one of two investigators (BP or JD), in a small meat ball of food, unless dietary requirements required dry pilling. If the latter occurred, water was administered by mouth after to promote transit of the pill to the stomach. The mouth was checked for any residual prednisolone after administration. The exact time of administration was recorded. All prednisolone was given in tablet form.

Dogs were housed in the veterinary hospital to which they were presented for the study, which was also their attending veterinary hospital. Dogs were housed individually and received standard nursing care.

The target sampling schedule included 10 1mL blood samples collected via intravenous catheter, including the baseline sample. Planned sample times were 0, 30, 60, 90, 120, 240, 300, 360, 420 and 480 minutes post prednisolone administration. Reasons for less than ten samples being collected were obstruction of the sampling catheter, dog owner schedule limitations, or other practical limitations on the day of the study. In addition, if the intravenous catheter that had been used for sampling occluded and was no longer viable, only two additional direct venepuncture samples were taken as set out in our ethic application.

Samples were collected into lithium heparin tubes and refrigerated until all samples were collected from the patient. Samples were centrifuged (1500 $g$, 20 minutes), the supernatant plasma transferred into Eppendorf tubes, and frozen at −80° C until analysis.

Three healthy dogs receiving no medications provided blood for the standard curve and quality control samples (Animal ethics ID 1814451.1, The University of Melbourne).

 

## Prednisolone quantification

Chemicals: Prednisolone and prednisolone-D6 were obtained from Sigma-Aldrich (Catalog number: 46656) and Santa Cruz Biotechnology (Catalog number: SC-219641) respectively. Prednisolone-D6 was selected as the internal standard (IS). Methanol, acetonitrile and formic acid were from Fisher Chemical, Merck and Ajax FineChem respectively. High-purity water was obtained from a Milli-Q apparatus.

Plasma sample preparation: Patient plasma (50 µL) was diluted with methanol (190 µL) and prednisolone-D6 internal standard solution (10 µL, 1.25 µg·mL$^{-1}$). The mixture was vortexed (15 s) and centrifuged (15,000 $g$, 15 min) and the supernatant (100 µL) was transferred to autosampler vials for LC-MS analysis.

Calibration sample preparation: Prednisolone and prednisolone-D6 stock solutions (1000 and 1.25 µg mL$^{-1}$ respectively) were prepared in methanol. Blank canine plasma (50 µL) was spiked with prednisolone-D6 (10 µL, 1.25 µg mL$^{-1}$) and diluted prednisolone stock to final prednisolone concentrations of 1, 3, 10, 50, 100, 250 and 500 ng mL$^{-1}$. Two quality control samples were prepared in the same way with prednisolone concentrations of 20 ng mL$^{-1}$, 200 ng mL$^{-1}$ and a prednisolone-D6 concentration of 50 ng mL$^{-1}$.

Liquid-chromatography tandem mass spectrometry: Samples were analysed using a Shimadzu 8050 triple quadrupole mass spectrometer coupled to a Shimadzu Nexera X2 liquid chromatography unit. Components of plasma, quality control or calibrations samples were separated using an Agilent Poroshell 120 EC-C18 column (2.1 x 50 mm, 2.7 µm) attached to an Agilent Poroshell SB-C18 guard column (2.1 X 5 mm, 2.7 µm) over an 8 min gradient with.1% formic acid in H$_2$O as mobile phase A and.1% formic acid in acetonitrile as mobile phase B. The solvent gradient was as follows [time (min), B (%)]: [0, 3], [2, 3], [5, 90], [6, 90], [6.5, 3], [8, 3]. The LC flow rate was.5 mL min$^{-1}$ and eluent was diverted to waste for the first 2 minutes of the gradient. The autosampler chamber and column oven were maintained at 5 °C and 40 °C respectively throughout the analysis.

Compounds eluting from the column were introduced into the gas phase by electrospray ionisation (ESI). The nebulising, heating gas flow rate was set to 2 L min$^{-1}$ and both the heating and during gas flow rates were set to 10 L min$^{-1}$. The interface temperature was set to 375 °C and the interface voltage was 4 kV. Prednisolone and Prednisolone-D6 were analysed in positive ion mode using a multiple reaction monitoring (MRM) strategy. Five transitions were acquired for each analyte S1 Table and collision energies and parameters were optimised using the in-built Shimadzu LabSolutions MRM optimisation tool.

LC-MS data analysis: Raw LC-MS data were loaded into Shimadzu LabSolutions Insight software. Chromatographic peaks at 3.95 and 4.00 assigned to prednisolone and prednisolone-D6 were detected and integrated using the in-built 'Chromatopac' algorithm with default parameters. Isotope ratios for two qualifiers were flagged in cases where the calculated value deviated by >20% from values calculated from calibration standard injections and, in these cases, chromatographic peak integrations were manually visualised and corrected if necessary.

Assay calibration and linearity: Calibration standards were analysed repeatedly (n = 4) on the first day of analysis and once on each of the subsequent two days. Seven-point calibration curves were constructed using weighted linear least-squares regression (1/x) of the ratio between prednisolone and IS peak areas vs the ratio of prednisolone to IS concentrations. The coefficient of determination ($R^2$) was used as the metric of assay linearity.

Precision and accuracy: Intraday precision and accuracy were determined by repeated analysis of quality control and calibration standards on the same day (n = 4). Interday precision and accuracy were assessed by analysis of calibration and quality control samples on each of the subsequent two days of data acquisition. Precision was calculated as the ratio of the standard deviation to the mean prednisolone concentration (i.e., the coefficient of variation, CV) and is expressed as a percentage. Accuracy was calculated as the average prednisolone concentration divided by the nominal concentration and expressed as a percentage.

Selectivity and carry-over: The selectivity of the assay was assessed by repeated analysis of prednisolone-free dog plasma that was prepared for analysis as per analytical samples, except that blank methanol (10 µL) was added instead of the prednisolone-D6 IS mixture. Blank plasma samples were analysed (n = 4) on the first day of analysis and at least once on each subsequent day. Data were manually inspected to verify the absence of analyte and internal standard

peaks. Method carry-over was assessed by injection and analysis of pure methanol in between each injection of calibration standard samples.

## Pharmacokinetic modelling

Data analysis was conducted in a two-stage approach using non-compartmental modelling. Estimates of the area-under-the-curve ($AUC_{8h}$) were obtained using a hierarchical generalized additive model (HGAM) [11,12]. This strategy was selected due to sparsity and imbalance in the observations, and the presence of censored observations which cannot be handled in standard workflow for non-compartmental analysis (NCA) [13]. The HGAM was implemented in R (Version 4.1.3 (2022)) [14] using package 'brms' [12,15]. From the completed HGAM, the estimated AUC for 8 hours after prednisolone administration ($AUC_{8h}$) for each subject was obtained by numeric integration.

The point estimates (posterior median) of $AUC_{8h}$ for each subject were passed forward as data to the second stage. The second stage model was a nonlinear regression model, which expressed $AUC_{8h}$ as a function of the dose and bodyweight. Apparent clearance $\frac{Cl}{F}$ was based the allometric model:

$$Cl/F = A \cdot BW^B \tag{2}$$

Conveniently this equation can be linearized:

$$\log_e \left( \frac{Cl}{F} \right) = \log_e (A) + (B \cdot \log_e (BW)) \tag{3}$$

In linear pharmacokinetics the AUC can be predicted from the clearance and the dose:

$$AUC_\infty \sim dose \cdot F/Cl \tag{4}$$

Substituting Equation 3:

$$AUC_\infty \sim dose/e^{\log_e (A)+(B \cdot \log_e (BW))} \tag{5}$$

Finally, under the presumption that $AUC_{8h}$ was a reasonable approximation of $AUC_\infty$, this nonlinear model was implemented as a regression model for the $\log_e(AUC_{8h})$, and specified in 'brms' syntax as:

```
bf(log(AUC_8hr) ~
log(DOSE_ng/(exp((thetaA)+(exp(thetaB)*(log(WEIGHT)-
log(20))))))), thetaA + thetaB ~ 1, nl = TRUE), …
```

with the subject bodyweight in kilograms WEIGHT, the dose in nanograms DOSE_ng, and the response variable AUC_8hr (ng·min·mL$^{-1}$) on the log-scale. This model has two primary parameters, the coefficient $A$ and exponent $B$ which are both implemented on log-scale, and two parameters associated with the Student-$t$ error model, the scale $\sigma$ and degrees-of-freedom $\nu$. The unknown parameters $A$ and $B$ characterize the relationship between prednisolone dose, bodyweight, and $AUC_{8h}$. The process of analysis is to learn about the values of these parameters from the data.

Estimation for the second-stage model was implemented with a Bayesian approach, selected for its ease of interpretation, and the opportunity to specify relevant prior knowledge [16]. This method prior information, in the form of a probability distribution, to be specified for each of the parameters to be estimated. The prior distribution for the exponent $B$, representing the key estimate for allometric scaling, was $N(-.288, .3)$ on the log-scale. On the linear scale, its median is about.75, the theoretical expectation for scaling hepatic function to bodyweight, and about 95% of its area is between.46 and 1.22. Prior information for the scaling coefficient $A$ was set by centering the bodyweight variable so that the parameter

represented the apparent clearance (mL·min⁻¹) for a 20 kg dog (as in the model expression above); the prior distribution for log(A) was $N(4.5, .75)$, which has median about 90 on the linear scale, and about 95% of its area between about 20 and 400. In the results the parameter is represented on non-centered scale consistent with Equation 3. Priors for the error model parameters used the package defaults. Principles and practice regarding prior information for veterinary pharmacology applications are described by the author [16].

Goodness of fit of the model was assessed by residual analysis and Bayesian $R^2$ [17]. Results were summarized using the posterior distributions of the primary parameters, and predictions from the model of the expected $AUC_{8h}$ for various bodyweight and dose combinations, including both bodyweight and BSA dosing. Visualizations were generated from these predictions, via 'ggplot2' and 'ggdist' [18].

## Results

### Subjects

Twenty-six dogs were enrolled in the study, with 25 contributing to data analysis; one dog, with multicentric lymphoma, was excluded due to having markedly delayed prednisolone detection in its blood samples; there was no prednisolone detected until 6 hours and 8 minutes after administration, which was the last sample collected for this dog, so its data were judged unsuitable for statistical modelling.

Key characteristics for the 25 dogs are described in Table 1 and additional individual information in S1 File. The median weight was 19.9 kg, ranging from 5.4 kg to 44.6 kg. There were 12 male neutered, 11 female spayed, 1 male entire and 1 female entire. There was a variety of breeds, with poodles and poodle crosses (n = 4) and Cocker spaniels (n = 3) being the only duplicate breeds. Chronic enteropathy (n = 5) and immune mediated thrombocytopenia (n = 4) were the most common indication for prednisolone administration. Dogs were receiving a variety of other medications including immunosuppressants (cyclosporine n = 4, leflunomide n = 2, chlorambucil n = 1), mineralocorticoids (desoxycorticosterone pivalate n = 3), antibiotics, analgesics, anti-thrombotics and anti-anxiety medications. There was one known MDR1 mutant. The median number of samples collected was 8. A total of 16 dogs had 8 or more samples including two dogs with 10 samples S1 File 1. For the majority of dogs, less than ten samples were collected when the sampling catheter became non-patent and only two additional direct venepunctures were obtained as approved by our ethics committee. In a few cases, dog owner schedule limitations, or other practical imitations on the day of the study prevented to obtain all samples.

### Prednisolone assay development

Assay calibration and linearity: The calibration curve for prednisolone was linear over the range 1 ng·mL⁻¹–500 ng·mL⁻¹ and the coefficient of determination was 0.9987. The coefficients of the regression line $y = cx + d$ were: $c = 1.74018$ and $d = 0.00693$.

**Table 1. Demographic variables of the 25 included dogs.**

| Variable | Median (range) | Range |
|---|---|---|
| Age (months) | 76 | 13–152 |
| Weight (kg) | 19.9 | 5.4–44.6 |
| Body condition score (/9) | 6 | 3–7 |
| Muscle condition score (/3) | 3 | 2–3 |
| Dose of prednisolone (mg) | 7.5 | 2–40 |
| Dose of prednisolone (mg/kg) | .44 | .08–2.20 |
| Number of samples collected | 8 | 4–10 |

Precision and accuracy: Intra- and inter-day precision and accuracy data for the determination of prednisolone concentrations in dog plasma are given in S2 Table. The intra-day accuracy ranged from 91.5% to 116.69% and the precision was between 2.2% and 6.4%. Inter-day accuracy ranged from 87.8 to 106.8% and precision varied between.6 and 11.2%. In each case, these results were within internationally accepted tolerances of 20% for the LOQ and 15% for all other standards [19].

Lower limit of quantitation: Prednisolone was detected and quantified with acceptable precision and accuracy (within 20%) at the lowest nonzero calibration point measured (1 ng·mL$^{-1}$).The LLOQ was less than.05% of the maximal prednisolone concentration previously measured in dog plasma by Nam *et al* [8] suggesting sufficient sensitivity.

Selectivity and carry-over: Manual inspection of chromatographic data from the analysis of plasma drawn from dogs not exposed to prednisolone revealed no significant interfering peaks in the region of retention time of 3.95 min (i.e., the expected retention time of prednisolone). Inspection of data from pure methanol injections following the analysis of calibration samples showed no quantifiable peaks for calibrators in the range of 1−250 ng·mL$^{-1}$. Following the injection of the 500 ng·mL$^{-1}$ standard, a minor peak was observed in subsequent analyses of pure methanol of approximately 15% the height of the 1 ng·mL-1 (i.e., LLOQ) calibration samples. These data are within the FDA-recommended tolerances of <20% carryover at the LLOQ [19].

## Prednisolone pharmacokinetics in dog plasma

Prednisolone plasma concentration-time relationships are presented in Fig 1. Dog 4 was excluded from data analysis due to delayed absorption and therefore not included in this figure. The lower limit of detection was 1ng/mL; most observations at time 0, and a handful of subsequent observations, were below the detection limit. Time of the maximum observed prednisolone concentrations after oral administration was highly variable, ranging from 24 min to 6h 44 min after administration. Four dogs had higher prednisolone concentrations at their last sample, than the sample prior, suggesting incomplete observation of $AUC_{8h}$.

Overall, the HGAM provided a close fit to the concentration data, including for subjects with few observations (Fig 1). The estimated concentration-time functions highlighted the erratic form of the data for some subjects, both in the initial and terminal phases, which was exacerbated in some cases by insufficient duration of observation. Estimated $AUC_{8h}$ ranged from about 9,750–229,000 ng·min·mL$^{-1}$; the $AUC_{8h}$ for each subject with its dose and bodyweight are expressed in S1 Fig. The observed apparent clearance, obtained as $\frac{Cl}{F} = \frac{dose}{AUC_{8h}}$, ranged from 3.1 to 55.0 mL·min$^{-1}$·kg$^{-1}$ S1 Fig.

The fit of the non-linear regression model for $AUC_{8h}$ based on dose and bodyweight was apparently reasonable based on residual analysis and posterior predictive check. The final model explained about 74.9% (90% CrI: 72.8–75.3%) of the variability (Bayes R$^2$) [17] in prednisolone $AUC_{8h}$. The point estimate (posterior median) of the allometric coefficient $A$, on the linear scale, was 22.8 (Fig 2). The point estimate (posterior median) of the allometric exponent $B$ was.83. Parameter estimates and their uncertainty are illustrated in Table 2.

The scaling exponent $B$ controls the shape of the relationship between $BW$ and $\frac{Cl}{F}$ dashed lines (from left) highlight $B = 2/3$, corresponding to the scaling function implied by body-surface-area-based dose determination, and $B = 1$ for bodyweight-based dose determination.

The scaling coefficient $A$ controls the vertical position of the relationship; its value corresponds to the expected $Cl/F$ (mL·min$^{-1}$) for a dog of bodyweight 1 kg.

The prior probability distributions are selected by the investigator to represent existing knowledge of the parameters. The posterior distributions represent knowledge after analysis has been conducted, taking into account the prior information, the model, and the data. In this case the overall evidence obtained from the data was modest.

Hypothesis testing for two important point values of $B$ of clinical interest, 2/3 and 1 (corresponding to body surface area and body weight dosing, respectively, which are highlighted in Fig 2, was conducted to summarize the extent of evidence contributed by the study. Hypothesis testing under a Bayesian approach may be conducted via the Bayes factor (BF) that

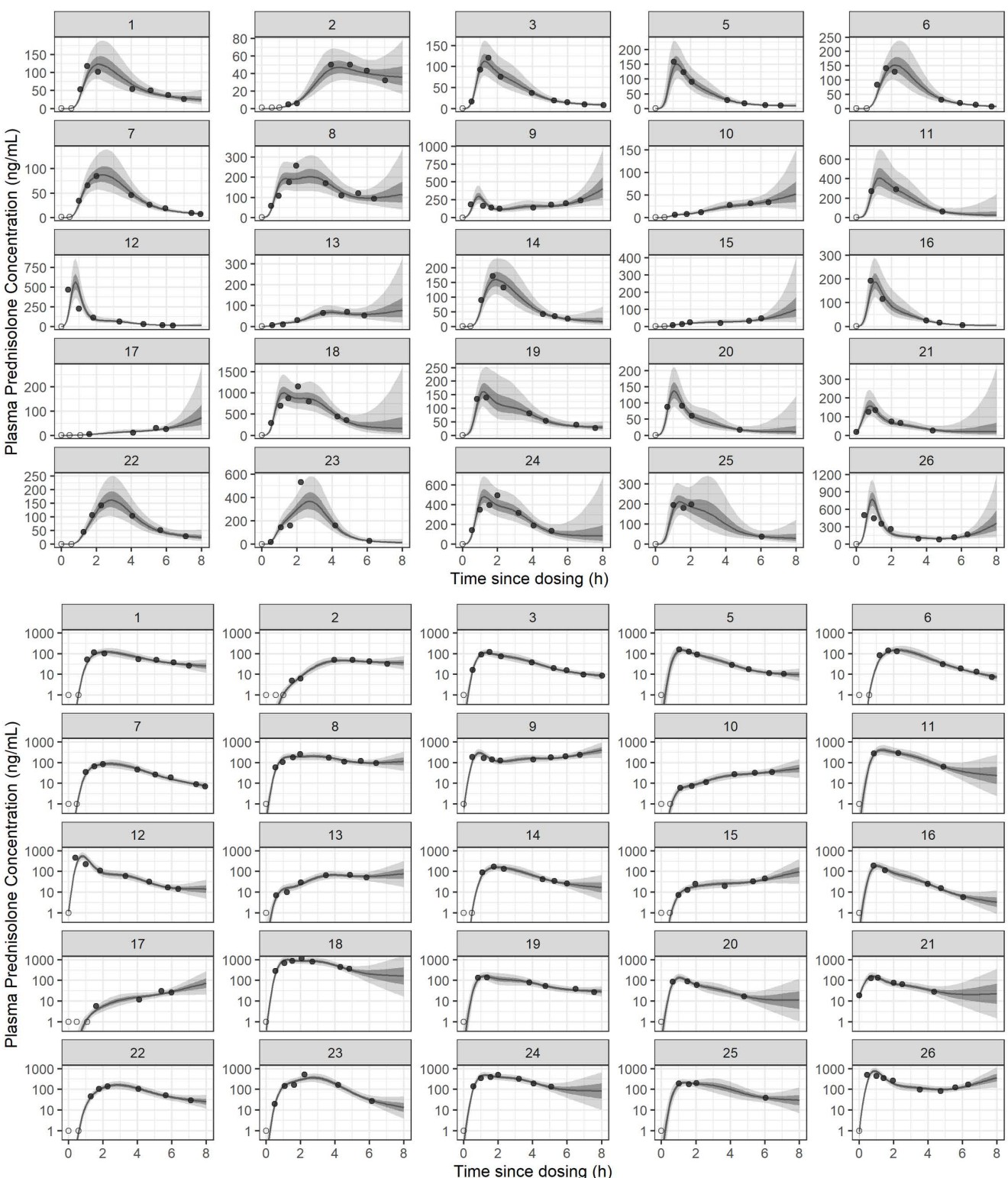

**Fig 1. Concentration-time relationships for prednisolone after single oral administration in 25 dogs (data from 1 additional subject were excluded), of varying dose.** Upper panels are on linear scale (varying y-axis range), and lower panels are semilogarithmic ($log_{10}$ y-axis). Filled points

are the observed prednisolone concentration and open points the censored observations (below the quantification limit). The solid line represents the posterior median predicted concentration from a semiparametric (smooth) model. The light grey field represents the 90% credible interval for the predicted concentration, and the dark grey field the 50% credible interval. The estimated $AUC_{sh}$ for each subject was obtained by integration of the posterior median concentration.

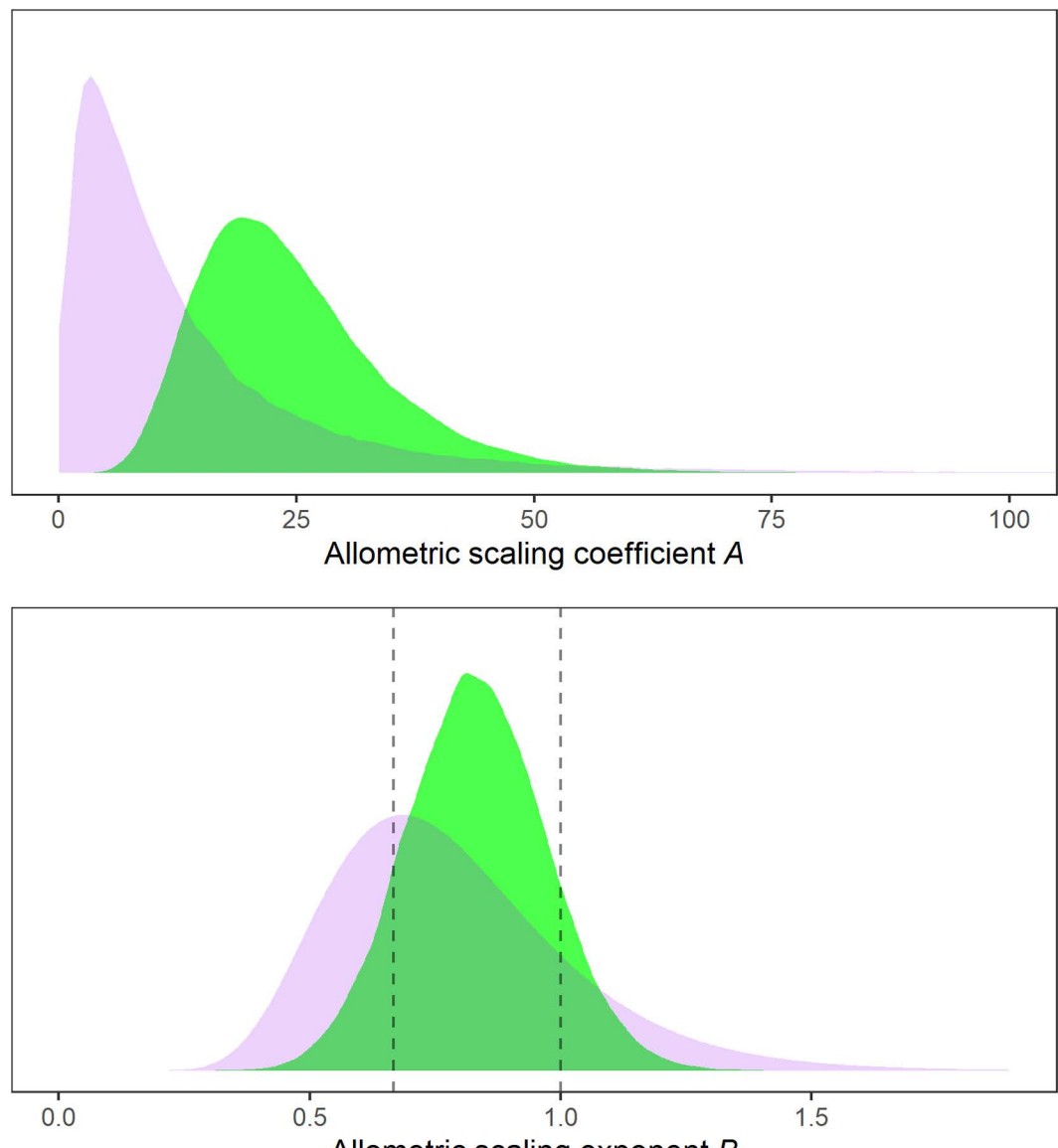

**Fig 2. Prior (purple) and posterior (green) probability distributions, for the parameters of the allometric scaling model for prednisolone apparent clearance $\frac{Cl}{F}$ (mL·min-1) after oral administration, as a function of bodyweight $BW$ (kg): $\frac{Cl}{F} = A \cdot BW^B$.**

**Table 2. Parameter estimates for the allometric model for the effect of bodyweight on $AUC_{8h}$.**

| Parameter | Posterior median | 5% CrL | 95% CrL |
|---|---|---|---|
| $\log_1 (A)$ | 3.13 | 2.46 | 3.78 |
| $A$ | 22.8 | 11.6 | 43.6 |
| $\log_1 (B)$ | -.19 | −.50 | .06 |
| $B$ | .83 | .60 | 1.06 |
| $\sigma$ | .47 | .33 | .64 |
| $\nu$ | 12.6 | 3.18 | 41.4 |

CrL: Limits of the credible interval (posterior quantiles).

$A$: Allometric scaling coefficient.

$B$: Allometric scaling exponent.

$\sigma$: Scale parameter for residual error.

$\nu$: Degrees of freedom parameter for residual error.

summarizes the evidence supporting one model over another. The Bayes factor for $B = 2/3$ was .81, and for $B = 1$ was 1.64. Though interpretation is subjective, and the use of thresholds generally discouraged, both indicate weak ('anecdotal' strength) evidence [20].

Plots describing model-predicted $AUC_{8h}$ as a function of dose (mg/kg or mg/m$^2$) in hypothetical dogs of varying bodyweight are presented in Fig 3.

The solid line is the posterior median predicted $AUC_{8h}$. The light grey field is the 90% credible interval for the population prediction, and the dark grey field the 50% credible interval. Intervals represent the degree of uncertainty in the true position of the relationship, based on the information available from the data. The dashed line is at 1 mg/kg (upper panels) or 20 mg/m$^2$ (lower panels).

## Discussion

This study to our knowledge is the first to systematically assess the effect of bodyweight on prednisolone pharmacokinetics in dogs treated for different clinical conditions. Most pharmacokinetic studies in companion animals are conducted in small numbers of homogenous subjects under experimental conditions. Instead, we opted for an observational design, in patient dogs already being treated with prednisolone. We obtained weak evidence regarding the relationship between bodyweight, dose, and $AUC_{8h}$ in dogs of 5.4 kg to 44.6 kg bodyweight. Posterior median .83 was larger than the theoretical .75 for the relationship between bodyweight and metabolic rate [9]. However, the relatively wide posterior distribution indicates that the current data were not sufficient to confidently refute that the relationship was simply linear, and in fact, the data provided minor evidence in support of this proposition (BF = 1.64). Overall, the results are most consistent with scaling exponent $B$ between 2/3 and 1; the posterior probability that $B$ was between 2/3 and 1 was about .77, compared to the prior probability of about .48 (Fig 2). Though a meaningful improvement in knowledge, at least under this prior information, the results clearly are not precise enough for clinical application, which is neatly illustrated by uncertainty regions in Fig 3. However, the observational approach utilizing clinical patients is promising, and extending this design with a larger sample size would be warranted to generate a predictive model suitable for dose individualization.

In linear pharmacokinetics AUC is a function of dose, oral bioavailability and clearance [21]; $AUC_\infty = \frac{dose \cdot F}{Cl}$. Dose selection as a linear function of bodyweight, as typical in small animal practice, makes the implicit assumption that clearance scales linearly in bodyweight, *i.e.,* with $B = 1$ [22]. In contrast, interspecies evaluation has demonstrated fairly consistently that pharmacokinetic processes are generally nonlinear in bodyweight, as predicted by the nonlinearity of the underlying physiology [23]. Empirical assessment remains critical as PK processes, especially as typically conceptualized,

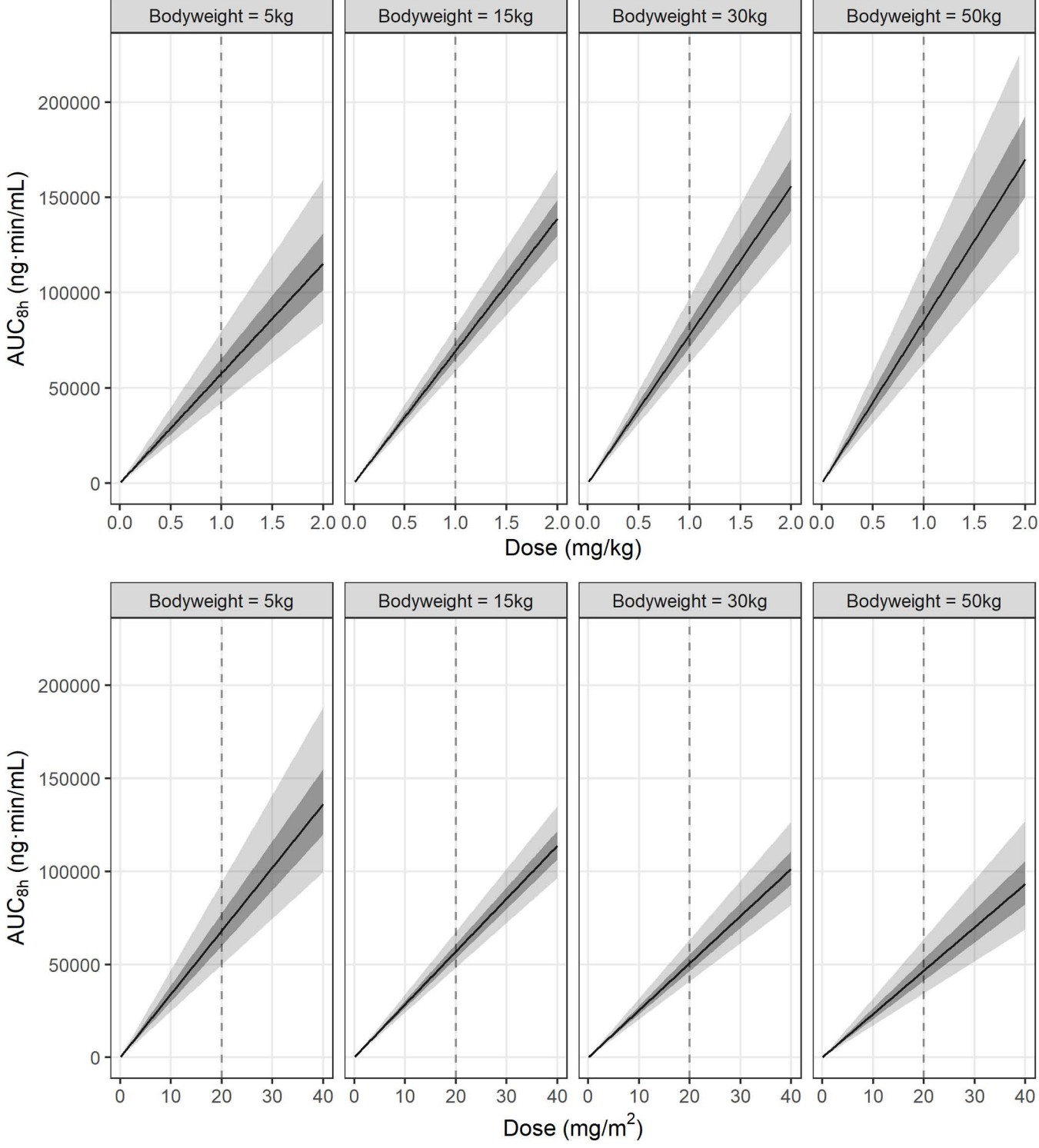

**Fig 3. Posterior-predicted area under the curve at 8 hours ($AUC_{8h}$) by dose, for a hypothetical average dog of 5 kg, 15 kg, 30 kg or 50 kg bodyweight.** The upper panels express the dose as a function of bodyweight (mg/kg), and the lower panels express the dose as a function of body surface area (mg/m²), with body surface area defined as $0.101 \cdot BW^{2/3}$.

are only semiphysiologic. For example, interspecies comparisons of dexamethasone clearance have estimated scaling with bodyweight $B$ of about.92 [24], while in human children prednisolone clearance was reasonably predicted by BSA [25].

The observed concentration-time relationship for some subjects in this study was consistent with erratic or delayed drug absorption (Fig 1). Previous studies under laboratory conditions describe a mean $T_{MAX}$ for prednisolone of.7–1.13 hours [8,26], but our observations suggest this was not consistently predictive for this cohort. Gastric emptying may be prolonged in dogs in the hospital environment, perhaps secondary to stress [27]. Similarly, absorption of digoxin was slower in hospitalised dogs [28]. Though our subjects were of generally good temperament, stress and delayed gastric emptying are a plausible explanation for these observations. Further, feeding status prior to admission was not uniform, so the presence of any residual food in the stomach could have influenced emptying. Regardless, these observations may simply reflect the typical extent of between-subject or between-occasion variability in the absorption pattern of prednisolone, so cannot be taken as artefacts. The opportunity to observe these effects is a meaningful external validity benefit of the observational study design.

The primary limitation of this study was incomplete observation of AUC in several dogs, which can be observed in Fig 1. Their $AUC_{8h}$ will be a meaningful underestimate of $AUC_\infty$ so their apparent clearance was overestimated. This is probably a relevant contributor to the large range of observed apparent clearance $\frac{Cl}{F}$, including observations as large as 55.3 mL·min$^{-1}$·kg$^{-1}$, which are likely too large to represent plasma clearance [29]. Though it was presumed in design that the 8h observation period would be sufficient, based on previous descriptions of prednisolone PK, under these conditions later observations were in fact required. Future observational studies evaluating oral prednisolone in dogs must consider exaggerated absorption delay and design accordingly. In principle, fully parametric statistical analysis, as in typical nonlinear multilevel modelling for population pharmacokinetics [30], could provide meaningful extrapolation for incompletely observed subjects and implement bodyweight as a nonlinear covariate. In this case, the parametric 'population' model was deliberately excluded due to the primary interest in drug exposure, and the expectation that a consistent absorption model would be statistically intractable; nonetheless, the novel HGAM-based approach realized many of its benefits, including partial pooling, correct implementation of censoring, and uncertainty quantification.

There was a meaningful degree of variation in $AUC_{8h}$ that was not described by dose or bodyweight. Without intravenous administration bioavailability $F$ is unknown, so between and within-subject variability in $F$ may be an important contributor, considering previous estimates of prednisolone bioavailability of 48–108% [26,31]. This would not contribute bias to our estimates if $F$ is not systematically related to bodyweight. Intravenous administration would be useful to reduce this unexplained variation, especially if combined with oral administration in the same subjects, but intravenous preparations of prednisolone are not in wide clinical use and were not available commercially at the time of this study. However, if $F$ is actually meaningfully variable in practice, it is an important contributor to the contextual relevance of the systematic effect of bodyweight; without good knowledge of not only the average $F$, but also its between and within-subject variation, the practical impact of varying clearance is difficult to determine.

Various patient factors may be relevant in this design. Ketoconazole and rifampicin have both been found to alter prednisolone AUC in dogs, likely due to alterations in P-glycoprotein expression [32], so dogs receiving these drugs were excluded. Other drugs administered in our sample population, such as cyclosporine, have the potential to alter P-glycoprotein expression [33], which may contribute to unexplained variability. The impact of disease on prednisolone pharmacokinetics in dogs is also unknown. Several dogs in this sample had intestinal disease which may alter oral bioavailability [34,35], although a recent study suggests that prednisolone absorption is similar between dogs with significant chronic intestinal diseae (protein losing enteropathy) and healthy dogs [36]. Bodyweight varies not only due to body size, but also body condition, and the meaningful variability in body condition in our subjects is an additional complication compared to more uniform laboratory animals. Though these and probably other factors contribute additional uncertainty in this study design relative to more experimental PK studies, these subjects are likely more representative of actual

veterinary patients, so benefits of the design are potential improvements in external validity, and an arguably lower ethical burden as subjects are studied in a natural state.

This study has provided a meaningful step towards dose personalization of prednisolone in dogs. Though some evidence was obtained regarding an allometric scaling model for prednisolone based on dose and bodyweight, the current data were not sufficient to provide substantial evidence against linear scaling of clearance with bodyweight, and the resulting parameter estimates were too uncertain to support direct predictions for clinical usage. A larger study, with a longer observation period, is warranted both to improve precision of the systematic effects, and to further quantify variability in prednisolone exposure.

## Supporting information

**S1 Fig. Prednisolone and Weight Scatter.**
(SVG)

**S1 Table. LCMS Peaks.** Optimised MRM parameters for Prednisolone and Prednisolone-D6. In the 'Product m/z' column, 'Q', 'R1' and 'R2' indicate the quantifier, and first and second qualifier ions respectively.
(DOCX)

**S2 Table. Intraday and interday precision and accuracy for the determination of prednisolone concentrations in Dog Plasma.**
(DOCX)

**S1 File. Dog information and prednisolone concentrations measured.**
(XLSX)

## Acknowledgments

The authors wish to thank Ted Whittem and Elizabeth Tudor for their advice on the study design and Babak Jalilian for his assistance in preliminary development of the liquid-chromatography tandem mass spectrometry method. The authors also wish to thank Lydia Hambrook and Sarah Helmond for their assistance in recruiting study participants.

## Author contributions

**Conceptualization:** Andrew P. Woodward, Julien Rodolphe Samuel Dandrieux.

**Formal analysis:** Andrew P. Woodward, Michael G. Leeming.

**Investigation:** Bonnie L. Purcell, Michael G. Leeming, Julien Rodolphe Samuel Dandrieux.

**Project administration:** Bonnie L. Purcell.

**Supervision:** Michael G. Leeming, Julien Rodolphe Samuel Dandrieux.

**Writing – original draft:** Bonnie L. Purcell, Andrew P. Woodward, Michael G. Leeming.

**Writing – review & editing:** Andrew P. Woodward, Julien Rodolphe Samuel Dandrieux.

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
