## [Decision Letter · Decision Letter 0]

Dear Dr. Dandrieux,

Thank you for submitting your manuscript to PLOS ONE. After careful consideration, we feel that it has merit but does not fully meet PLOS ONE’s publication criteria as it currently stands. Therefore, we invite you to submit a revised version of the manuscript that addresses the points raised during the review process.

We look forward to receiving your revised manuscript.

Kind regards,

Mark Zabel

Academic Editor

PLOS ONE

Journal Requirements:

American College of Veterinary Internal Medicine Resident Research Grant

Reviewers' comments:

Reviewer's Responses to Questions

**Comments to the Author**

1. Is the manuscript technically sound, and do the data support the conclusions?

Reviewer #1: Yes

Reviewer #2: Partly

2. Has the statistical analysis been performed appropriately and rigorously?

Reviewer #1: Yes

Reviewer #2: Yes

3. Have the authors made all data underlying the findings in their manuscript fully available?

Reviewer #1: Yes

Reviewer #2: No

4. Is the manuscript presented in an intelligible fashion and written in standard English?

Reviewer #1: Yes

Reviewer #2: Yes

Reviewer #1: This study is relevant to current veterinary practices and seeks to answer a question regarding prednisolone dosing in dogs that has long been a source of debate. The pharmacokinetics were analyzed appropriately and the study design was valid although as the authors point out a longer sampling time would have been useful due to the variability of absorption. The authors comment on variation in body condition score and I would be very curious to see the pharmacokinetic data with respect to body condition score in addition to weight. I would recommend that the authors insert a table or preferably figure with the BCS of the animal as well as the AUC as prepared in Figure 1. Although a firm recommendation on dosing is not possible from this study it is a good start to answer an important physiologic question.

Reviewer #2: The authors present an interesting study on the effects of bodyweight based prednisolone dosing in dogs. It is indeed unfortunate that no real conclusions can be generated other than discussing the wide variations in pharmacokinetics between dogs.

From looking at the plasma concentrations in Figure 1, it appears that there are 3 populations of dogs here: The classical early peak followed by a long declining tail, a biphasic group, and a group that never reaches peak concentrations in the sampled interval. Did you separate these out to see if there were any defining factors that could explain this variation - drugs, GI disease, etc. I have concerns about trying to model PK when there are such obvious differences in bioavailability and absorption patterns between subjects.

In the abstract (line 37) you note that a non-linear regression model 'described the relationship', but you weren't actually able to effectively model the relationship between BW and plasma concentrations. Did you apply a model, or attempt to model, or other? Perhaps just a language 'thing', but it reads to me like you were successful in fully modeling the relationship. Also in the abstract, the coefficients in the mathematical expression are lower case, but the description of them uses upper case.

You spend a fair amount of time discussing BSA in the introduction, but none of your aims involve this term. Did you set out to determine if BSA based dosing was a better method? Or are you just using this as an example of allometric scaling? I also don't see specifically where you modeled both dose in mg/kg and dose in mg/m2?

Line 89 - what did you use as criteria for 'clinically apparent liver disease'?

Lines 88 and 101 - One of the criteria for inclusion was that the dogs were amenable to intravenous catheter placement, then you say that you placed an intravenous catheter, but later on you note that some dogs did not get an intravenous catheter (line 116). Which was it?

Did you see if there were differences in PK for those that received the pill in for vs those that didn't (line 106)

Line 114 - How many of the patients got all 10 samples collected. You note that many of them did not for various reasons, but not how many actually completed the study.

In your quantification section, I would suggest referring to your methods as 'liquid-chromatography tandem mass spectrometry (LC-MS/MS)' as in the abstract for consistency (and technical correctness).

I am no mathematician, so I could be wrong on this, but you seem to have separated the terms on the right side of Equation 2 using both log base 10 and natural log together. Is this technically correct?

Line 238 - you note that only 25 of the 26 dogs contributed to data analysis, but there 26 curves in Figure 1, and in fact the legend for Figure 1 indicates that you are presenting data from 25 dogs.

**Do you want your identity to be public for this peer review?** For information about this choice, including consent withdrawal, please see our Privacy Policy

Reviewer #1: **Yes: ** Valerie Johnson

Reviewer #2: **Yes: ** Gregg M Griffenhagen

---

## [Author Response · Author response to Decision Letter 1]

23 Apr 2025

Reviewers' comments:

Reviewer's Responses to Questions

Comments to the Author

Reviewer #1: This study is relevant to current veterinary practices and seeks to answer a question regarding prednisolone dosing in dogs that has long been a source of debate. The pharmacokinetics were analyzed appropriately and the study design was valid although as the authors point out a longer sampling time would have been useful due to the variability of absorption. The authors comment on variation in body condition score and I would be very curious to see the pharmacokinetic data with respect to body condition score in addition to weight. I would recommend that the authors insert a table or preferably figure with the BCS of the animal as well as the AUC as prepared in Figure 1. Although a firm recommendation on dosing is not possible from this study it is a good start to answer an important physiologic question.

Thank you for the positive feedback to our manuscript and the suggestion of inclusion of the body condition score to our data.

From the physiologic perspective the relevance of the body condition score would be in the relationship between body weight and some notion of ‘body size’, with the hypothesis that the scaling of organ function is directly with body size rather than body weight. This may explain some of the remaining between-subject variation in our analysis, as the bodyweight is an imperfect surrogate for the true predictor ‘body size’.

Although this is reasonable in principle, it would be statistically challenging for various reasons. Particularly important would be the rigor of body condition scoring measurement process; as the measure is subjective and generally vague in the clinical usage, careful prospective planning and justification would be necessary for meaningful results. The high potential for collinearity of the body weight and body condition score suggests that even under good measurement conditions an exaggerated sample size would be needed to tease out the direct effects simultaneously, and this may be difficult to predict in advance, especially if balance in the distribution of body weight and BCS couldn’t be assured.

This is especially true in our current set of dogs, with 19 out of 25 dogs having a BCS of 5 or 6 (S1. File)

We feel that it is doubtful that the data at hand would provide meaningful information about BCS due to these limitations. For this reason, we would respectfully suggest not including such a figure.

Reviewer #2: The authors present an interesting study on the effects of bodyweight based prednisolone dosing in dogs. It is indeed unfortunate that no real conclusions can be generated other than discussing the wide variations in pharmacokinetics between dogs.

From looking at the plasma concentrations in Figure 1, it appears that there are 3 populations of dogs here: The classical early peak followed by a long declining tail, a biphasic group, and a group that never reaches peak concentrations in the sampled interval. Did you separate these out to see if there were any defining factors that could explain this variation - drugs, GI disease, etc.

Thank you for your thoughtful review of our manuscript. Dogs with chronic enteropathy (including protein-losing enteropathy) had a classical absorption. This is consistent with a recent study from Dr Jablonski (DOI: 10.1111/jvim.17277) that reported no difference in PK between dogs with PLE and healthy dogs.

There are some major statistical problems with attempting to explain emergent patterns in the data after-the-fact. This is essentially a form of HARKing. Because the study was not designed with these effects in mind there is a substantial risk that these types of associations have no causal interpretation, and also that any such comparisons lack much power. This suggests that any patterns observed through this sort of reasoning will be unstable and will not generalize. This process also embeds researcher confirmation bias, because the observed effect (the data pattern) and the proposed explanation emerged from the same data. It is reasonable to suggest, based on a priori reasoning, what effects might have contributed, but not to try and assess those based on these data. For these reasons, our preference is not to further separate the groups, but this should be kept in mind for future studies.

I have concerns about trying to model PK when there are such obvious differences in bioavailability and absorption patterns between subjects.

This concern presumes that variation in bioavailability and absorption presents some validity threat. I’m not sure why this should be the case. This perspective would generally preclude inference from pharmacokinetic data except under closely controlled experimental conditions, which defeats the whole notion of population PK. Here, we chose to conduct the inference using a noncompartmental approach, which is invariant to the absorption process. Though the incompleteness of observation of some subjects does limit the accuracy of the final analysis, it is not clear why this should preclude doing statistical inference altogether.

It is important to keep in mind that if there is large between or within-subject variation in bioavailability under field conditions, this has a strong influence on the contextual relevance of the systematic effect of bodyweight on exposure (in statistical terms, the ‘effect size’). The opportunity to assess that to some extent is actually a relative advantage of this design compared to working with experimental animals as in most veterinary PK studies, which artificially constrain variation.

In the abstract (line 37) you note that a non-linear regression model 'described the relationship', but you weren't actually able to effectively model the relationship between BW and plasma concentrations. Did you apply a model, or attempt to model, or other? Perhaps just a language 'thing', but it reads to me like you were successful in fully modelling the relationship. Also in the abstract, the coefficients in the mathematical expression are lower case, but the description of them uses upper case.

We agree that this is likely predominantly an issue of terminology. We have attempted to be as clear and transparent as possible about the modelling workflow throughout the manuscript. I’m not sure what is meant by the phrases ‘apply a model’ or ‘attempt to model’. The methodology specifies exactly what models were generated and how they were generated, and their findings are presented in multiple forms in the results.

The phrase in the abstract ‘described the relationship’ is simply an articulation of what was done. It conveys no claim about how much was learned from doing so or that anything of interest was demonstrated or proved.

The completed nonlinear regression model which described AUC as a function of bodyweight and dose captured a meaningful amount of the total variability (Bayes R^2: 74.9). The extent to which this is ‘effective’ is entirely subjective, but it is objectively the case that the degree of variation described by the model was large. That the available data did not contain sufficient evidence to decisively support the value of key parameters is a different consideration.

There is no standard usage of the term ‘fully modelling’. The statistical model includes features that capture systematic effects of the predictors of interest, and remaining unexplained randomness. These are ‘full’ in the sense that they provide a description of the data that is apparently internally valid, and addressed the questions of interest.

Thank you for noticing the discrepancy in the coefficients. We prefer the upper case, so have revised them for consistency throughout.

You spend a fair amount of time discussing BSA in the introduction, but none of your aims involve this term. Did you set out to determine if BSA based dosing was a better method? Or are you just using this as an example of allometric scaling? I also don't see specifically where you modeled both dose in mg/kg and dose in mg/m2?

The common dosage scaling as a ratio of body weight ‘per kilogram’ assumes a simple linear relationship between dose and exposure, i.e. B = 1. Scaling of dosage by body surface area is a popular alternative in clinical applications, especially in oncology, and assumes B = 2/3. It has been recently suggested to use body surface area when dosing prednisolone for large dogs diagnosed with immune-mediated haemolytic anaemia (DOI: 10.1111/jvim.15463) as they apparently developed more fequent adverse effects from prednisolone.

These are two special cases for the general allometric model. Under the hypothesis that larger dogs experience relative overdosing because B<1, it would be clinically convenient, and biologically interesting, if B = 2/3 were an accurate approximation (the data at hand suggest that it is not). A potential misinterpretation of this system, that is important to address, is that ‘bodyweight’ and ‘body surface area’ dosing represent discrete choices, but actually each are just specific examples of an underlying continuous model, and neither may apply.

Body surface area can be obtained as a simple allometric function of body weight (kg), using the expression 0.101× 〖BW〗^(2/3), as expressed in the manuscript. So, any prednisolone dosage expressed as mg/kg can be expressed equivalently as mg/m2, and vis-versa, by simple conversion, if body weight is known. The statistical model describes AUC as a function of dosage ‘mg’ and bodyweight ‘kg’, so the generating predictions regarding dosage in mg/kg and mg/m2 was done simply by manipulating the input variables. No separate statistical modelling is required, and would be inappropriate because such models would be implicitly closely related and not separately interpretable.

Line 89 - what did you use as criteria for 'clinically apparent liver disease'?

Thank you for this comment, dogs had either no evidence of synthetic failure or normal bile acid stimulation test (e.g. rule out of significant liver disease, for example for the dog with protein-losing enteropathy). The following has been added to clarify this further:

Line 89: (no evidence of synthetic failure or normal bile acid stimulation test)

Lines 88 and 101 - One of the criteria for inclusion was that the dogs were amenable to intravenous catheter placement, then you say that you placed an intravenous catheter, but later on you note that some dogs did not get an intravenous catheter (line 116). Which was it?

Thank you for highlighting this discrepancy. We had ethics set out in case not all dogs were amenable to catheter placement, but they actually were (samples collected are listed in S. file 1). We have removed line 116

Did you see if there were differences in PK for those that received the pill in for vs those that didn't (line 106).

Most dogs were given the pill in a small meat ball or with a piece of chicken. No pattern was noted and although it is reasonable to pose this as a potential contributor, we would not be able to assess this further with the current dataset.

Line 114 - How many of the patients got all 10 samples collected. You note that many of them did not for various reasons, but not how many actually completed the study.

This has been clarified in the manuscript (line 254):

“A total of 16 dogs had 8 or more samples including two dogs with 10 samples S1 File 1. For the majority of dogs, less than ten samples were collected when the sampling catheter became non-patent and only two additional direct venepunctures were obtained as approved by our ethics committee. In a few instances, dog owner schedule limitations, or other practical imitations on the day of the study prevented to obtain all samples.”

The data are fully expressed for the reader in S1 file as well as figure 1, along with the goodness-of-fit of the accompanying model, which provides the reader with the ability to see exactly what data were being used and how the number and distribution of observations for each subject affects the results.

In your quantification section, I would suggest referring to your methods as 'liquid-chromatography tandem mass spectrometry (LC-MS/MS)' as in the abstract for consistency (and technical correctness).

Thank you for this comment, we have adjusted the manuscript for consistency as suggested.

I am no mathematician, so I could be wrong on this, but you seem to have separated the terms on the right side of Equation 2 using both log base 10 and natural log together. Is this technically correct?

Thank you for noticing this mistake. You are right, there is actually no usage of the base 10 logarithm anywhere in the model. In equation 3 there was a typographic error and the character e was missed; this has been corrected in the updated submission.

Line 238 - you note that only 25 of the 26 dogs contributed to data analysis, but there 26 curves in Figure 1, and in fact the legend for Figure 1 indicates that you are presenting data from 25 dogs.

Figure 1 includes only 25 curves. The following sentence has been added in the manuscript to clarify (Line 287):

“Dog 4 was excluded from data analysis due to delayed absorption and therefore not included in this figure.”

---

## [Decision Letter · Decision Letter 1]

Influence of bodyweight on prednisolone pharmacokinetics in dogs

PONE-D-25-03648R1

Dear Dr. Dandrieux,

We’re pleased to inform you that your manuscript has been judged scientifically suitable for publication and will be formally accepted for publication once it meets all outstanding technical requirements.

Kind regards,

Mark Zabel

Academic Editor

PLOS ONE

Additional Editor Comments (optional):

Reviewers' comments:

Reviewer's Responses to Questions

**Comments to the Author**

Reviewer #1: All comments have been addressed

2. Is the manuscript technically sound, and do the data support the conclusions?

Reviewer #1: Yes

3. Has the statistical analysis been performed appropriately and rigorously?

Reviewer #1: Yes

4. Have the authors made all data underlying the findings in their manuscript fully available?

Reviewer #1: Yes

5. Is the manuscript presented in an intelligible fashion and written in standard English?

Reviewer #1: Yes

Reviewer #1: The authors have adequately addressed all concerns and although results are ambiguous I support their decision to not try to make claims when there is inadequate evidence to do so.

**Do you want your identity to be public for this peer review?** For information about this choice, including consent withdrawal, please see our Privacy Policy

Reviewer #1: **Yes: ** Valerie Johnson

---

## [Editor Report · Acceptance letter]

PONE-D-25-03648R1

PLOS ONE

Dear Dr. Dandrieux,

I'm pleased to inform you that your manuscript has been deemed suitable for publication in PLOS ONE. Congratulations! Your manuscript is now being handed over to our production team.

Kind regards,

on behalf of

Dr. Mark Zabel

Academic Editor

PLOS ONE